# Chiral Magnetic Josephson Junction as a Base for Low-Noise Superconducting Qubits

**Maxim N. Chernodub** [1,*] , **Julien Garaud** [1] **and Dmitri E. Kharzeev** [2,3,4]

1 Institut Denis Poisson CNRS/UMR 7013, Université de Tours, 37200 Tours, France
2 Department of Physics and Astronomy, Stony Brook University, New York, NY 11794-3800, USA
3 Department of Physics and RIKEN-BNL Research Center, Brookhaven National Laboratory, Upton, New York, NY 11973, USA
4 Le Studium, Loire Valley Institute for Advanced Studies, 1 Rue Dupanloup, 45000 Orléans, France
* Correspondence: maxim.chernodub@univ-tours.fr

**Abstract:** The lack of space inversion symmetry endows non-centrosymmetric superconducting materials with various interesting parity-breaking phenomena, including the anomalous Josephson effect. Our paper considers a Josephson junction of two non-centrosymmetric superconductors connected by a uniaxial ferromagnet. We show that this "Chiral Magnetic Josephson junction" (CMJ junction) exhibits a direct analog of the Chiral Magnetic Effect, which has already been observed in Weyl and Dirac semimetals. We suggest that the CMJ can serve as an element of a qubit with a Hamiltonian tunable by the ferromagnet's magnetization. The CMJ junction avoids using an offset magnetic flux in inductively shunted qubits, thus enabling a simpler and more robust architecture. Furthermore, when the uniaxial ferromagnet's easy axis is directed across the junction, the resulting "chiral magnetic qubit" provides robust protection from the noise caused by magnetization fluctuations.

**Keywords:** qubit; Josephson junction; topological semimetal; chiral magnetic effect

## 1. Introduction

The discovery of superconductors lacking spatial inversion symmetry [1–5] has opened the possibility of studying the spontaneous breaking of continuous symmetry in a parity-violating material. In particular, the superconducting order parameter in these non-centrosymmetric superconductors (NCSs) is a parity-odd quantity [4,5], enabling several interesting magnetoelectric phenomena due to the mixing of singlet and triplet superconducting parameters, correlations between supercurrents and spin polarization, the appearance of helical states, and the peculiar structure of Abrikosov vortices (see [6,7] for a review). Notably, in these superconducting materials, vortices can show an inversion of the magnetic field away from the vortex core [8,9] and new compact states that are the superconducting counterparts of the Chandrasekhar–Kendall states in highly conducting plasmas [10].

Parity breaking in NCSs also results in an unconventional Josephson effect, where the junction features a phase-shifted current relation [11,12]:

$$J(\varphi, \varphi_g) = J_c \sin(\varphi - \varphi_g). \tag{1}$$

Here, $\varphi$ is the superconducting phase difference across the junction, $J_c$ is the critical Josephson current, and $\varphi_g$ is the parity-breaking phase offset. Nonzero bias $\varphi_g \neq 0$ results in a nonvanishing current across the junction, even when the phase difference $\varphi$ is zero. Since the current is a parity-odd quantity, this signals a parity violation.

Phase-biased junctions (often called "$\varphi_0$-junctions") have been suggested to appear in a wide range of systems, including non-centrosymmetric [11,13,14] and multilayered [15] ferromagnetic links between conventional superconductors, topological insulators [16,17],

nanowires [18,19], quantum point contacts [20], quantum dots [21–23], and Weyl semimetals [24,25]. The first experimental realization of Josephson $\varphi_0$-junctions has been reported in superconductor–quantum dot structures, where the phase offset $\varphi_g$ is controlled via electrostatic gating [26].

In this article, we introduce the Josephson junction made of two NCSs weakly linked by a uniaxial ferromagnet with an easy axis normal to the interface, i.e., parallel to the electric current (see Figure 1). Unlike in previous proposals [11,27,28], the ferromagnetic exchange field $h$ here is directed normally to the NCS/F/NCS interfaces. Parity breaking in NCS couples the magnetization $h$ to the supercurrent $j$, resulting in a term $j \cdot h$ in the Ginzburg–Landau free-energy functional describing crystal structures with $O$ point group symmetry. As derived below, this results in a nonzero current even in the absence of phase gradients across the junction. This current, directed along the magnetic field, stems from the breaking of parity in a non-equilibrium state and is thus a direct analog of the Chiral Magnetic Effect [29] predicted for systems of chiral fermions and observed in Dirac and Weyl semimetals [30–33]. This analogy motivates our terminology "Chiral Magnetic Josephson junction" (CMJ junction) to describe the NCS/F/NCS junction displayed in Figure 1. Below, we demonstrate that the current across the CMJ junction is still given by the expression (1), where the magnitude of the bias $\varphi_g$ can be tuned by the ferromagnet's magnetization.

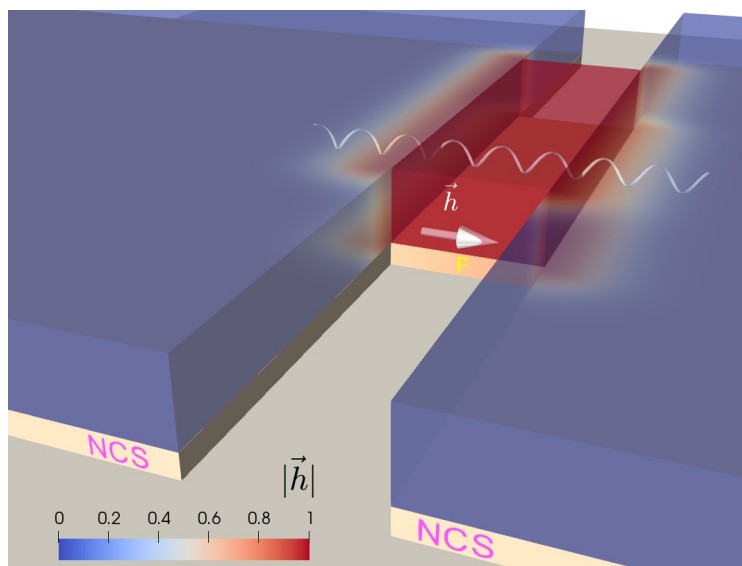

**Figure 1.** The Chiral Magnetic Josephson junction: two non-centrosymmetric superconductors (NCSs) are weakly linked by a uniaxial ferromagnet (F). The exchange field $h$ of the ferromagnet, oriented across the link, induces an inversion symmetry-breaking component of the supercurrent (represented by the spiral) in the junction.

We propose to use the CMJ junction as a constituent of a superconducting qubit. The junction's energy associated with the current (1) is

$$E(\varphi, \varphi_g) = E_J[1 - \cos(\varphi - \varphi_g)], \qquad (2)$$

where $E_J$ is the Josephson energy. The total energy of the qubit $E_Q$ is the sum of the junction's energy (2) and a term quadratic in the phase difference $\varphi$. For example, in the case of an inductively shunted junction [34], this quadratic term results from the inductive energy $E_L$:

$$E_Q(\varphi, \varphi_g) = E_J[1 - \cos(\varphi - \varphi_g)] + E_L\varphi^2. \qquad (3)$$

Here, the offset $\varphi_g$ plays the role of an offset flux and can be used to control the form of the qubit Hamiltonian. Using the magnetization of the ferromagnetic link should simplify the qubit architecture by avoiding the use of an offset flux and the corresponding source.

Noise in the offset flux is an essential component of qubit decoherence [35,36]. As demonstrated below, the noise in the offset phase $\varphi_g$ results from the fluctuations of the component of the magnetization normal to the interface. In the proposed setup, this direction corresponds to the easy axis of the uniaxial ferromagnet. Thus, only longitudinal magnetization fluctuations contribute to the noise, but these are suppressed by the ratio of the qubit temperature to the Curie temperature of the ferromagnet, which is about $10^{-5} - 10^{-4}$. Moreover, the current in the CMJ junction is parallel to magnetization and thus is "force-free", i.e., not subjected to a Lorentz force. This dramatically reduces the coupling between the current and the magnetization that contributes to the noise.

The offset phase $\varphi_g$ of the CMJ junction can be estimated within the Ginzburg–Landau (GL) framework. The superconducting state in non-centrosymmetric superconductors is commonly believed to be a mixture of singlet and triplet pseudo-spin states [6,7] due to the spin–orbital coupling in the presence of the broken inversion symmetry [37]. Using $\hbar = c = 1$, the Ginzburg–Landau free energy describing the superconducting state of non-centrosymmetric material reads as [7,38]:

$$f = a|\psi|^2 + \gamma|D\psi|^2 + \frac{b}{2}|\psi|^4 + \frac{K}{2}\boldsymbol{j} \cdot \boldsymbol{h}. \tag{4}$$

The single-component superconducting order parameter $\psi = |\psi|e^{i\varphi}$ is coupled to the vector potential $\boldsymbol{A}$ of the magnetic field $\boldsymbol{h} = \boldsymbol{\nabla} \times \boldsymbol{A}$ via the gauge derivative $\boldsymbol{D} = -i\boldsymbol{\nabla} - 2e\boldsymbol{A}$, while the coefficients $b$, $\gamma$ and $a = \alpha(T - T_c)$ are standard phenomenological GL parameters. The parity-odd nature of the non-centrosymmetric superconductor is reflected by the last term (Lifshitz invariant) of the free energy (4), which describes the direct coupling of the magnetic field $\boldsymbol{h}$ to the usual, parity-odd component of the supercurrent density:

$$\boldsymbol{j} \equiv \boldsymbol{j}^{\mathrm{odd}} = 2e\gamma[\psi^*\boldsymbol{D}\psi + \psi(\boldsymbol{D}\psi)^*]. \tag{5}$$

Note that the exchange field $\boldsymbol{h}$ of the ferromagnet plays the role of the background magnetic field $\boldsymbol{B}$. The parity-odd last term in (4) yields an additional parity-even contribution to the total supercurrent $\boldsymbol{J}$:

$$\boldsymbol{J} = \boldsymbol{j}^{\mathrm{odd}} + \boldsymbol{j}^{\mathrm{even}}, \qquad \boldsymbol{j}^{\mathrm{even}} = 4e^2\gamma K|\psi|^2\boldsymbol{h}. \tag{6}$$

The GL functional (4) describes NCS materials with $O$ point symmetry, such as $Li_2Pt_3B$ [5,39] and $Mo_3Al_2C$ [40,41], and the coupling constant $K$ determines the magnitude of the superconducting magnetoelectric effects following the broken inversion symmetry. Our derivation applies to non-centrosymmetric superconductors with other crystallographic groups with a generic Lifshitz invariant $K_{\alpha\beta}h_\alpha j_\beta$. In this case, when the $x$-axis is directed across the normal link, the diagonal element $K_{xx}$ should be nonzero. Notice that Lifshitz invariants of the type $\boldsymbol{n} \cdot \boldsymbol{h} \times \boldsymbol{j}$ do not have such a diagonal element and thus cannot satisfy this requirement [1].

As illustrated in Figure 1, we consider a pair of identical non-centrosymmetric superconductors separated by a uniaxial ferromagnetic weak link whose internal exchange field $\boldsymbol{h} \equiv h_x\boldsymbol{e}_x$ points across the link. We neglect the term quartic in the condensate and disregard inhomogeneities of both the condensate $\psi$ and the exchange field $\boldsymbol{h}$ in the transverse $yz$-plane. The minimization of the GL free energy (4) with respect to the superconducting order parameter in the background of the ferromagnetic exchange field $\boldsymbol{h}$ then yields the equation

$$a\psi - \gamma\frac{\partial^2\psi}{\partial x^2} - 2ie\gamma Kh_x\frac{\partial\psi}{\partial x} = 0 \tag{7}$$

that describes the tunneling of the Cooper pairs across the weak link. For the time being, we assume the absence of an external electromagnetic field at the link, $A = 0$. Due to proximity effects, the tunneling of the Cooper pairs between the non-centrosymmetric superconductors through the centrosymmetric weak link will not respect the parity inversion $x \to -x$, as can be seen from Equation (7).

The general solution of Equation (7) for the superconducting gap inside the weak link reads as:

$$\psi(x) = C_+ e^{q_+ x} + C_- e^{q_- x}, \tag{8}$$

where the wavevectors

$$q_\pm = \pm \sqrt{\frac{a}{\gamma} - (eh_x K)^2} - ieh_x K \tag{9}$$

should have a nonzero real part so that the weak link is in a normal state, thus requiring $a > a_c = \gamma (eh_x K)^2$. The coefficients $C_\pm$ in Equation (8) are determined by the boundary conditions at the interfaces of the weak ferromagnetic link with the superconductors at $x = \pm L/2$. It is customary to make a simplification using rigid boundary conditions [11,42], which assume the absence of a barrier at the interfaces and imply continuity of the superconducting order parameter:

$$\psi(x = \pm L/2) = |\Delta| e^{\pm i\varphi/2}; \tag{10}$$

here, $|\Delta|$ is the absolute value of the order parameter at the superconducting leads.

Using the relation Equations (8)–(10) together with the definition of the total current (6) yields the phase-shifted current relation:

$$J = J_0 \sin(\varphi - \varphi_g), \quad J_0 = \frac{4e\gamma |\Delta|^2 \sqrt{\frac{a}{\gamma} - (eh_x K)^2}}{\sinh L \sqrt{\frac{a}{\gamma} - (eh_x K)^2}}, \tag{11}$$

which exhibits the offset of the phase difference given by

$$\varphi_g = eh_x KL. \tag{12}$$

This offset, corresponding to the parity-breaking phenomenon, is proportional to the strength of the magnetic interaction $K$ in the Ginzburg–Landau free energy (4). In other words, the presence of the nonzero phase bias $\varphi_g \neq 0$ signals the breaking of the parity-inversion symmetry between leftward and rightward tunneling of the Cooper pairs and leads to a nonzero current in the "steady state" of the junction even if the phase difference between the superconducting leads is zero, $\varphi = 0$.

In a long junction, $L\sqrt{a/\gamma - (eh_x K)^2} \gg 1$, the current (11) is an exponentially small quantity due to suppression of the Cooper-pair tunneling between widely separated superconducting leads. The limit of a short junction in the presence of parity breaking should be taken with care. In the thermodynamic equilibrium $\varphi = \varphi_g(L)$, the current through the junction is always zero. However, in the steady state with zero phase difference $\varphi = 0$, the electric current does not vanish and is given by the parity-even term (6):

$$\boldsymbol{J}(\varphi = 0, L \to 0) = 4e^2 \gamma K |\Delta|^2 \boldsymbol{h}. \tag{13}$$

This current plays a crucial role in the dynamics of the chiral magnetic qubit. As mentioned earlier, the current (13) shares a striking similarity with the Chiral Magnetic Effect [29], with $\boldsymbol{h}$ and $K$, respectively, playing the roles of the external magnetic field and the source of the parity breaking.

Some features of the phenomenon that we discuss also appear in usual centrosymmetric *s*-wave superconductors separated by the Josephson junction made of an NCS-type

ferromagnet with *tangentially*-oriented field $h$. Even though the underlying dynamics are quite different, the latter system is described by an equation similar to (7) [11]. However, in our case, the currents that flow along the magnetic field are force-free and, thus, are not subjected to the noise resulting from the transverse fluctuations of magnetization.

Similar types of Josephson junctions with the ferromagnetic exchange field $h$ oriented transversally between two non-centrosymmetric superconductors with the $C_{4v}$ point group (corresponding to interactions of the type $n \cdot h \times j$) have been proposed in [27,28]. The Josephson current in a non-ferromagnetic junction between two non-centrosymmetric superconductors does not exhibit the offset in the phase [43]. A finite phase offset for the Josephson current of chiral charge appears, however, for the junction between two Weyl superconductors separated by a Weyl semimetal and for a magnetic field oriented transversally [44].

Notice that an ordinary Josephson junction driven by an external AC electric current possesses an $I$–$V$ curve with a ladder structure characterized by regions with constant voltage. These regions, known as Shapiro steps, are precisely specified by parameters of the Josephson junction, allowing for accurate frequency–voltage conversion in the experimental setting [45]. For noncentrosymmetric superconductors, the positions of the Shapiro steps should be shifted due to the magnetoelectric effect, which contributes to the voltage drop in the presence of the ferromagnetic exchange field $h$. The phenomenon is similar to the Josephson junctions involving the topological insulator as the weak link sandwiched between two type-2 superconductors. In such a device, the magnetoelectric effect is caused by axion coupling $E \cdot B$ in the topological insulator that breaks the time-reversal symmetry of the system and affects Shapiro steps [46].

We estimate the phase bias (12) numerically as follows:

$$\varphi_g \simeq 1.5 \times 10^{-3} \, h(\mathrm{T}) \, L(\mathrm{nm}) \, K(\mathrm{nm}). \tag{14}$$

The length of the ferromagnetic Josephson junction is typically of the order of tens of nanometers ($L \sim 30$ nm in [47]). The exchange field $h$ should not exceed the upper critical field $H_{c2}$, which, for several NCS superconductors, may reach significant values, $H_{c2} \sim 10$ T. Fields of this order and higher are known to be created by usual ferromagnets [48]. Note that even for magnetic fields larger than $H_{c1}$, the vortex formation can be avoided by choosing a weak link with a sufficiently small cross-section such that the total magnetic flux entering the superconductor is smaller than the flux quantum $\Phi_0$. The main uncertainty in our estimate comes from the poorly known parity-odd coupling $K$. Its value was estimated to be $K \simeq (10^{-3} \ldots 10^{-2})\lambda$ [49,50], where $\lambda \simeq (0.1 \ldots 1)$ $\mu$m is the penetration depth. Despite this uncertainty, the phase bias may be tuned to take values of order $\varphi_g \sim \pi$. For a given NCS superconductor, the phase bias can be manipulated by the magnetization of the weak link.

The chiral magnetic Josephson junction sketched in Figure 1 can be inductively shunted, for example, by a series of conventional Josephson junctions, to form a "chiral magnetic qubit" (see Figure 2). Such circuits include, in addition, two mixed Josephson junctions between the conventional and NCS superconductors. These mixed junctions do not generate the electric current across them at zero phase difference $\varphi = 0$ [51].

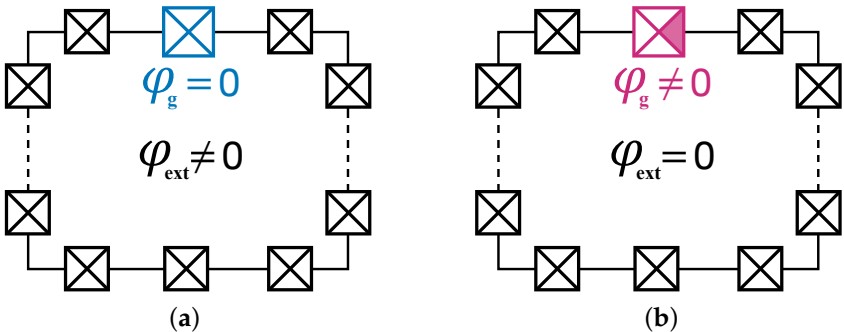

**Figure 2.** (**a**) Fluxonium-type qubit based on conventional Josephson junction inductively shunted by a series of Josephson junctions. The qubit is biased by an external magnetic flux $\varphi_{\mathrm{ext}} \equiv 2\pi\Phi/\Phi_0$, where $\Phi_0$ is the elementary flux quantum. The gate phase offset for a conventional Josephson junction is absent, $\varphi_g = 0$. (**b**) Chiral magnetic qubits based on the Chiral Magnetic Josephson (CMJ) junction are inductively shunted by a series of Josephson junctions. The CMJ junction possesses an internal phase offset $\varphi_g \neq 0$ eliminating the need for an external magnetic flux $\varphi_{\mathrm{ext}}$.

The Coulomb interactions between the Cooper pairs are described by the kinetic term in the Hamiltonian of the qubit:

$$\hat{H} = 4E_C\hat{n}^2 + E_J[1 - \cos(\varphi - \varphi_g)] + E_L\varphi^2, \tag{15}$$

where $\hat{n} = -i\hbar\partial_\varphi$ is the Cooper-pair number operator, and the last two terms describe the Josephson tunneling and the induction (3). The Hamiltonian (15) is generic for a family of inductively shunted qubits, including the fluxonium [52,53], one-junction flux qubits, and flux-biased phase qubits [54].

As illustrated in Figure 2a, fluxonium qubits relate the phase offset to the externally applied flux $\Phi$ as $\varphi_g = 2\pi\Phi/\Phi_0$, where $\Phi_0 = h/(2e)$ is the flux quantum. These are further characterized by a specific set of model parameters, such as the small inductive energy ($E_L/E_J \simeq 0.045$) and a moderate charging energy ($E_C/E_J \lesssim 1$), which give a unique combination of long coherence time and large anharmonicity of the energy levels [55]. Transmon qubits, on the other hand, are characterized by Coulomb charging energy much smaller than the Josephson tunneling energy, $E_C \ll E_J$, thus reducing noise caused by the offset charge fluctuations [56].

A nonzero phase bias $\varphi_g \neq 0$ imposes a large anharmonicity on the energy-level structure [34] determined by the Schrödinger equation:

$$\hat{H}\psi_n(\varphi) = \varepsilon_n\psi_n(\varphi). \tag{16}$$

The regime $\varphi_g = \pi/2$ provides maximum level-splitting and the absence of nearly degenerate level pairs [34]. Figure 3 displays the structure of the energy levels corresponding to this Hamiltonian. The transitions between the first excited state and the ground state, $|1\rangle \to |0\rangle$, can be substantially suppressed by the barrier separating them. This barrier is almost absent in the typical fluxonium regime ($E_L = 0.045E_J$ and $E_C = E_J$), as the first excited energy level $\varepsilon_1$ practically coincides with the height of the barrier, Figure 3a. This conclusion is valid to good accuracy for a wide range of values of the phase offset $\varphi_g$.

Decreasing the Coulomb energy towards the transmon regime leads to the appearance of the prohibitive barrier for transitions $|1\rangle \to |0\rangle$ between the different wells, Figure 3b. As displayed in Figure 3c, further decreases in the Coulomb energy reduce the energy difference between $|1\rangle$ and $|0\rangle$ states. The lifetime of the first excited level may be enhanced by lowering the inductive energy $E_L$. Figure 3d shows that at $E_L = 0.01E_J$, the barrier is sufficiently high to ensure quasi-classical protection of the first excited level. Related discussions of the energy levels can be found in [55,57] for fluxonium-type qubits.

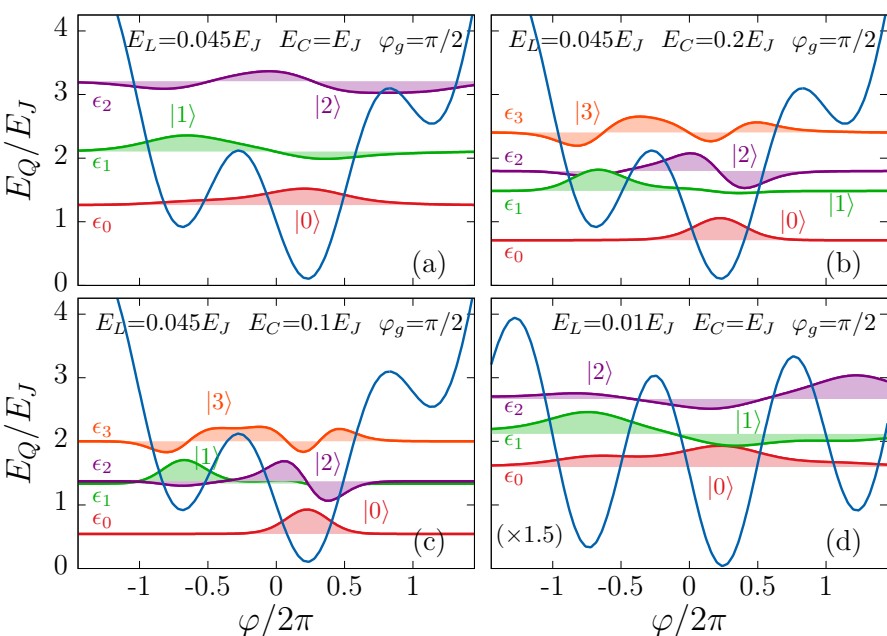

**Figure 3.** The potential energy (3) of the chiral magnetic qubit with the chiral magnetic Josephson junction possessing the phase offset $\varphi_g = \pi/2$ for various Coulomb charging energies $E_C$ and inductive energies $E_L$: (**a**) $E_C = 0.045E_J$, $E_L = E_J$; (**b**) $E_C = 0.045E_J$, $E_L = 0.2E_J$; (**c**) $E_C = 0.045E_J$, $E_L = 0.1E_J$; and (**d**) $E_C = 0.01E_J$, $E_L = E_J$. Lowest eigenstates $|n\rangle$ with $n = 0, 1, \ldots$ are shown along with the numerically computed energy levels $\varepsilon_n$ and the corresponding wavefunctions $\psi_n(\varphi)$.

The conventional way to induce the phase $\varphi_g$ in fluxonium qubits is to apply a background magnetic flux $\Phi$. The noise $\delta\Phi/\Phi_0$ in magnetic flux is typically of the order of $10^{-3} - 10^{-2}$; overcoming this noise is a central problem in quantum computer design. In our case, the noise in $\varphi_g$ is due to the noise in magnetization. Indeed, Equation (12) yields the noise relation

$$\left(\frac{\delta\varphi_g}{\varphi_g}\right) = eKL\left(\frac{\delta h_x}{h_x}\right). \tag{17}$$

To reduce the noise in magnetization, we propose to use a highly anisotropic uniaxial ferromagnet, e.g., of magnetoplumbite type. In the case of a uniaxial ferromagnet, the 3D rotational symmetry is explicitly broken by the crystalline lattice's symmetry. The only surviving symmetry is 2D rotations around the easy symmetry axis of the ferromagnet in the basal plane perpendicular to this axis. In this case, magnetization fluctuations are given by the simplified form of the Landau–Lifshitz–Gilbert equation with no gyroscopic term. They correspond to the rotation of magnetization around the easy axis (which here points along the $x$ axis) with fluctuating components of magnetization $h_y$ and $h_z$ but with a fixed $h_x$, which is an integral of motion. Since the phase offset $\varphi_g$ (12) depends only on $h_x$, the transverse fluctuations of magnetization do not induce noise in this quantity.

Unlike the transverse ones, the longitudinal fluctuations of magnetization (i.e., fluctuations of the magnitude of $h$) induce noise in the offset phase $\varphi_g$. However, longitudinal magnetization fluctuations are expected to be suppressed compared to the transverse ones by a factor of $c\,T/T_C$, where $T$ is temperature, $T_C$ is Curie temperature of the ferromagnet, and $c$ is a constant of order one. Indeed, the transverse fluctuations correspond to gapless Goldstone modes with kinetic energy $\sim T$, while the longitudinal one is massive with energy $\sim T_C$. Analysis [58,59] of the Landau–Lifshitz–Bloch equation (including both transverse and longitudinal fluctuations of magnetization) indicates that the constant $c \simeq 2/3$. Therefore, at temperatures of the superconducting qubits, which are on the order of tens of milli-Kelvin, with Curie temperatures on the order of 1000 K, we expect the suppression of longitudinal fluctuations by a factor of $\sim 10^{-4} - 10^{-5}$. This allows us to

expect a suppression in the noise resulting from the offset flux of the CMJ junction as compared to external flux noise by a significant factor of $10^{-2}$.

The domain structure in uniaxial ferromagnets is known to depend crucially on the anisotropy [60]. In weakly anisotropic ferromagnets (Landau–Lifshitz type), there is branching of domains close to the surface, and the magnetic flux does not leave the ferromagnet. This is not a desirable domain configuration, as the magnetic field has to penetrate the superconductor. On the other hand, in strongly anisotropic ferromagnets (Kittel type), such as magnetoplumbite, the domains do not branch, and thus the magnetic flux does escape the ferromagnet. The superconducting interface may affect the domain structure of a thin ferromagnetic film [61]; this question requires further investigation.

To summarize, we introduced a Josephson junction consisting of two non-centrosymmetric superconductors connected by a uniaxial ferromagnet, and we demonstrated that it exhibits a direct analog of the Chiral Magnetic Effect. We proposed this Chiral Magnetic Josephson junction (CMJ junction) for use as an element of a qubit with parameters tunable by the ferromagnet's magnetization. The resulting Chiral Magnetic Qubit is protected from noise caused by fluctuations in magnetization and does not require an external magnetic flux, allowing for a simpler and more robust architecture. The main uncertainty stems from the poorly known parity—the odd response of non-centrosymmetric superconductors—and we believe that these materials and the properties of their interfaces deserve further study.

**Author Contributions:** Conceptualization, D.E.K.; methodology, M.N.C.; software, J.G.; formal analysis, J.G. and M.N.C.; investigation, D.E.K.; writing—original draft preparation, J.G., M.N.C. and D.E.K.; visualization, J.G.; supervision, D.E.K. All authors have read and agreed to the published version of the manuscript.

**Funding:** The work of D.K. was supported by the U.S. Department of Energy, Office of Nuclear Physics, under contracts DE-FG-88ER40388 and DE-AC02-98CH10886, and by the Office of Basic Energy Science under contract DE-SC-0017662.

**Data Availability Statement:** Not applicable.

**Conflicts of Interest:** The authors declare no conflict of interest.

## Abbreviations

The following abbreviations are used in this manuscript:

CMJ     Chiral Magnetic Josephson
CME     Chiral Magnetic Effect

## Note

[1]     Possible candidates for the NCS superconductors should thus have a crystalline structure with either the point group $O$ ($Li_2Pt_3B$, $Mo_3Al_2C$), $T$ point group (e.g. LaRhSi, LaIrSi), or $C_4$ ($La_5B_2C_6$), $C_2$ (UIr), etc. On the other hand, the point groups $C_{nv}$ with $n = 2, 3, 4, 6$ (possessed, for example, by the compounds $MoS_2$, MoN, GaN, $CePt_3Si$, $CeRhSi_3$, amd $CeIrSi_3$ [7]) correspond to the Lifshitz invariants of the type $\boldsymbol{n} \cdot \boldsymbol{h} \times \boldsymbol{j}$ that do not fit our proposal.

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
