# Peer review of "Chiral Magnetic Josephson Junction as a Base for Low-Noise Superconducting Qubits"

_universe, doi:10.3390/universe8120657_

Round 1
Reviewer 1 Report
The phenomenology of the Josephson effect proposed in this manuscript is simple and clearly shows the parallels between the Josephson effect proposed by the authors and the chiral magnetic effect. In this setup two non-centrosymmetric superconductors are connected by a uniaxial ferromagnetic tunnel junction. The fact that the magnetic exchange is along the direction of the current clearly emulates the chiral magnetic effect. I also liked the proposal of using this as a qubit.
In my opinion this manuscript is clearly written and can be published in Universe. Before I make my final recommendation, I would like to ask only one small question to the authors, that they could perhaps address in a couple of sentences. Some elements of this manuscript reminds me a discussion of a Josephson effect with a topological insulator junction having an external magnetic field applied along the junction (similarly to the exchange field in Fig. 1 of the manuscript), see PRL 117, 167002 (2016). In that paper the magnetoelectric effect causes an induced voltage and, consequently, leads to the appearence of Shapiro steps. What would be the authors expect regarding the Shapiro steps?
Author Response
We thank the Reviewer for the report. To address the concern raised, we included on page 5 of a revised manuscript a new paragraph that discusses the Shapiro steps in our context. We detail the changes introduced in the manuscript marked with the color.
Reviewer 2 Report
The authors of the manuscript propose a new type of parity-breaking Josephson tunnel junction, with a ferromagnetic magnetization h that is perpendicular to the normal layer separating the two superconducting regions. It results in a h dot j(-) term in the Ginzburg-Landau Gibbs free energy, Eq. 4, where j(-) denotes the parity-breaking odd part of the super current. The authors thereby predict a parity-breaking super current, Eq. 1. They further explore the dynamical predictions for a qubit based on such a parity-breaking Josephson tunnel junction in the second half of the paper. The authors emphasise that transverse noise is much reduced because the magnetization is perpendicular to the ferromagnetic layer that separates the two superconducting regions.
The above study is very interesting, and it merits publication. I noticed one thing, however, that the authors may want to address before publication. The h dot j(-) term that they add to the Ginzburg-Landau free energy, Eq. 4, predicts an "amperean" contribution to the super current that is proportional to curl j(-). This is due to the relationship h = curl A that the authors state there. What bearing does this have on the super current of the Josephson junction, Eq. 1 ? Does it result in the parity symmetry breaking described by the author's results encoded by Eqs. 11 and 12, or is it something else?
Author Response
First of all, we thank the Reviewer for the report.
Yes, the j·B coupling leads to the parity-breaking phenomena, which result in the appearance of the phase offset (12) of the superconducting condensate, which has a direct contribution to the Josephson current (1). We made this statement clearer in the revised manuscript (we detail the changes introduced in the manuscript marked with the color).
Reviewer 3 Report
The authors present an idea of a new type of superconducting qubit, namely a chiral magnetic qubit. Such a device does not require an external magnetic flux, which is an advantage of the approach.
The article is well-written and contains both the theoretical basis and numeric support of the idea.
I recommend the article for publication in its present form.
Author Response
We thank the Referee for the careful reading of our manuscript and the positive assessment of our work.